# The Dialectics of (Deep) Accessible Tourism and Reality—Hermeneutics of a Journey to Madrid

Jácint Farkas [1,2,3], Zoltán Raffay [4,*], József Kárpáti [5], Zsófia Fekete-Frojimovics [6] and Lóránt Dénes Dávid [5,7]

1  Department of Management, Faculty of Finance and Accountancy, Budapest Business School,
   H-1149 Budapest, Hungary
2  Research Centre for Astronomy and Earth Sciences—Eötvös Loránd Research Network (Excellent Research
   Centre of the Hungarian Academy of Sciences), H-1121 Budapest, Hungary
3  Institute for Advanced Studies, Business Ethics Center, Budapest Corvinus University,
   H-1093 Budapest, Hungary
4  Institute of Marketing and Tourism, Faculty of Business and Economics, University of Pécs,
   H-7622 Pécs, Hungary
5  Faculty of Economics and Business, John von Neumann University, HU-6000 Kecskemet, Hungary
6  Budapest Business School, Faculty of Commerce, Hospitality and Tourism, HU-1054 Budapest, Hungary
7  Institute of Rural Development and Sustainable Economy, Hungarian University of Agriculture and Life
   Sciences (MATE), H-2100 Godollo, Hungary
*  Correspondence: raffayz@ktk.pte.hu; Tel.: +36-20-9290723

**Abstract:** The authors have made an attempt in this case study, which is based on 'subjective' travel and existence experiences, for the indispensable separation of technical accessibility and fundamental or 'deep' accessibility—in both interpretation and application—and then to reconsider these concepts in their special philosophy-centred study, which is at the same time built on empirical inquiries and analyses. This is in line with a series of their publications in high-class periodicals. The authors are aware and understand at first sight that this hybrid analysis method has several shortcomings concerning objectivity expected by the academic community, and also concerning the verification of the findings with exact data. Nevertheless, they are convinced that in today's world of transdisciplinarity, subjective and objective viewpoints are no longer dimensions mutually excluding each other in research. Accordingly, the 'artificially' created boundaries between ontological and epistemological philosophical approaches are not of substantial character either. In fact, their very necessity and even their existence can be questioned at certain types of applications (e.g., hermeneutical and Buddhist analytics). The experiences gained and analyses made by the authors make it clear that technical accessibility, and the interpretation and implementation of fundamental accessibility, as well as the control of these by the actual users, are still hindered by several obstacles. Additionally, the existence or lack of fundamental accessibility is a more significant issue than the mere fact of providing accessibility by technical solutions. Last but not least, it should be remarked that it is just the spirit of fundamental accessibility and its implementation in the real world that is capable of mostly meeting the personal needs for accessibility, which seems to be partially impossible. The paper is hermeneutic in nature, so it seeks to understand and interpret a phenomenon, and not to causally explain something. Accordingly, the reported data (both subjective and objective facts) serve the purpose of hermeneutics and not that of providing empirical proof.

**Keywords:** accessibility; safety; understandability; philosophy; subjectivity

## 1. Introduction

First, the authors emphasise that the academic readership should note the following: their paper is an investigation with a special hermeneutic-dialectical focus, which moves in synchrony with the following quote from Feyerabend: "But 'subjectivity' and 'objectivity'

mix equally in both domains and the case of the two friends turns up everywhere. Indeed, without 'understanding' their equipment in the sense in which some historians claim they 'understand' a distant historical figure scientists would not get anywhere. Today we can say even more. The speculations connected with superstrings, twistors, alternative universes no longer consist in formulating assumptions and then testing them, they are much more like developing a language that satisfies certain very general constraints (though it need not satisfy them slavishly) and then constructing a convincing and beautiful story in terms of this language. It is really very much like writing a poem" [1] (p. 146).

Moreover, this subjectivity-centred approach can be brought close to the same platform as the authors' 'research methodology', i.e., the practical and theoretical modalities of Buddhist hermeneutics, which will be discussed later. It must also be pointed out that this typically unconventional mode of inquiry and textual practice, as Feyerabend repeatedly emphasizes, is also a fundamental characteristic of the work of the classical Greek philosophers (Socrates, Plato, Aristotle, etc.). That is, they did not think and train their disciples in abstract systems of general descriptions of reality, but rather circumscribed a particular theme or 'element of being' in terms of the actual manifestations of the present at the time. Therefore, the science vs. philosophy axis that appears in the quote, and is found in the writing as well, is only apparent. The aim is to identify the strengths and weaknesses of two qualitatively not dissimilar fields and to explore in part the mutually barrier-removing nature of the two approaches.

In light of these, the article is a special case study with the intention to raise interest, which is not unique in the world of academic publications [2,3]. The authors have set out by the publication of this paper—which is academic and philosophical at the same time and is based on a case study (To avoid misunderstandings, let us be specific: this is not to contrast the two approaches, but rather to illustrate the complementary non-hierarchical relationship between the two aspects of investigation in this context.)—to sketch the coordinate system of accessibility, safety, and understandability, and to demonstrate in this relationship that the need for accessibility and safety are not exclusively relevant to disabled travellers. One can just think of the general obstacles caused by the use of English language analysed in the paper. In order to make this essay as compatible with the academic mainstream as possible, the authors are presenting it in the form of a review paper.

The hermeneutical scrutiny method that the authors chose entails a special blend of Western and oriental (Buddhist) philosophical 'research methodology' [4]. Consequently, both the philosophical and academic) literature that the authors processed and the way they processed it represent a specific non-linear interpretation framework. The essence of this is that the authors and ideas cited organise the text corpus in themselves, i.e., in an implicit way the whole of the text contains its own literature review and the necessary explanations. It should be remarked that the hermeneutical interpretation method recommended by Gadamer and called organic by the authors of this article actually 'demands' that the readers relate to the respective piece of work, not only in a didactical way but intellectually uniting with that, also finding sovereign thoughts and correlations that have not been mapped before. This Gadamerian 'interpretation methodology' shows organic similarities with the already-mentioned Buddhist hermeneutical approach: "The following should be seen as an attempt to read the Greek classics of philosophy differently this time: not starting with the critical superiority of modernity, which modernity thinks that, possessing an endlessly sophisticated logics, it now surpasses all that is old, but on the firm belief that "philosophy" has remained an experience of the human being conflicting him with events, which dignifies him as man, and that there is no progress in it, there is only sharing. And the fact that this is true for a civilisation like ours, shaped by science—it sound incredible, and still it seems true to me." [5] (Translated from the Hungarian version).

The main characteristics of Buddhist hermeneutics, which the authors also use as a quasi-research method, are, without claiming completeness, the following. In its examination, definition, and understanding of a given 'existing entity' and its contexts, it explicitly avoids the linear and hierarchical approach, as well as the dualistic approach still used

in mainstream positivist methodologies [4]. It strives to encourage readers, investigators, and thinkers to continuously think and to develop a wide range of reflexions by 'jumping' between different sub-disciplines. It is important to stress that Buddhist hermeneutics is both theoretical and practical. However, it explicitly avoids single-person or individual 'investigations', knowing that human consciousness relates to and interprets reality in different ways. As a consequence, the aim is primarily to bring the 'contexts of reality' of the persons under investigation closer to each other. Last but not least, it seeks to maintain and demonstrate the positive aspects of alert consciousness not only in theory but also in practice. This way, 'meeting' the requirements of the original teaching of the Buddha, it also aims at the improvement of the ability to remember. As an illustration of this there will be, for example, a brief reflection made in the case study summary to the shock to tourism induced by the COVID-19 pandemic and the need to remember the given trauma both on an individual and community level. All this is briefly discussed in the conclusion chapter of the paper, thereby also pointing to the relevance of the aforementioned seemingly eclectic text corpus construction mode, which is both an indispensable component and practice of Buddhist hermeneutics. Just as it was in view of the practical guidelines of this 'methodology', five researchers, all with different perspectives, collaborated to compile this review paper. Thus, what the authors' community calls the incomplete, mosaic draft picture is intended to refer to the juxtaposition of different interpretative surfaces. Therefore, due to its dialectical and hermeneutic nature, and in order to identify problems and propose solutions, the specific case study written by the authors is primarily aimed at members of the scientific community, in a way untypical in the mainstream.

Closely related to this is the 'category choice' of the study. There are different types of case studies, in many cases partly overlapping, especially in the world of philosophy [1,5]. This inspired authors to choose and develop a very specific 'mix' of the given approaches. It has two components, one of which is the participatory approach to disability studies, in which the lived experiences of existence and life of the person(s) concerned are expressed in a specific, one might say, subjective way, and all of which also reflect aspects of interpretation that are significantly influenced by disability. The second component, closely related to this, is the practical application of Buddhist hermeneutics, whereby the 'practitioner' inevitably weaves his or her own world of experience into the picture of the interpretative horizon of the subject in question, and then the respective findings are interpreted in a dialectical, specific discussion by the community. In this case, the aim of the debate is not to 'defeat' one of the participants, but to find a common position on the interpretation of the dimension in question.

## 2. Philosophical and Scientific Background

Studies by the authors, published in 2022 in highly-ranked periodicals, aimed to identify the paradigmatic differences and identical aspects related to technical accessibility and fundamental (or 'deep') accessibility, and to define the existential dimensions of the two concepts [6–8]. Further common features of the papers cited are of a philosophical nature. The respected shared aspects pertain to the non-linear and hierarchical 'architecture', more typical of the oriental philosophies, whose application serves the purpose of complementing the 'didactical' and data-centred approach of the academic mainstream by an innovative attitude that provokes thoughts and promotes a mental attitude capable of adopting new ideas. It should nevertheless be remarked in relation to these philosophical trains of thoughts that the ontological and epistemological distinction between fundamental (deep) accessibility and technical accessibility, typical in the 'Western world', is not adequate. In the non-linear and non-hierarchical relations of the interpretations of the world and the conscience, dualities and distinctions provide easier understanding in the best case, but when understanding becomes real and experienced, the 'reality transcripts' of duality nature will automatically disappear [9]. Furthermore, one of the cornerstones of Heidegger's fundamental ontology is also centred around this statement: the expression and worldview called 'physis', used in Antique Greek philosophy, simply saw man as an

integral part of the cosmos and did not find it important to make the epistemological and ontological distinctions and demarcations [10].

All this said, technical accessibility can also be seen as an ontological factor; consequently, fundamental (deep) accessibility is also an ontological category.

The authors argue that technical accessibility and fundamental accessibility, from a scientific point of view, do not have an independent paradigm, although in the case of accessibility this has already been attempted by researchers in the field of disability studies [11]. In short, they believe that technical accessibility is a human activity with a technological focus, which aims to ensure accessibility and the independent use of the built environment and natural places within certain limits by transforming, rebuilding, and changing these areas.

Fundamental accessibility, in a different way, is mostly a set of human attitudes, which is our anthropological core property. In other words, in the authors' view, the evolutionary history of humans naturally includes the potential for adaptation, socialisation, mutual assistance and, not least, the transformation of the living environment, i.e., it is one of the intellectual drivers that make us humans [12–14].

In the authors' view, the ability to consider the two activities and concepts as separate paradigms in scientific terms is essential. In this case, in the dimension of travel science, the focus of investigation is limited to the technological coordinate system of technical accessibility, and thus a destination that seems to be near perfect and can be called barrier-free is included in the category of accessible [15].

This case study of a week-long trip to Madrid in the summer of 2022 seems to briefly resonate with the abovementioned criteria. Due to space constraints, the authors can only sketch an incomplete, mosaic draft picture for their readers and colleagues. They hope that the brief empirical research presented in the second part of the study and the theoretical framework of a philosophical nature that is built around it can be further expanded in the coming years and will serve as inspiration for professional discourses.

The authors are convinced that people who want to travel, whether they live with 'only' an existential or also a functional disability [7,16], arriving in a 'good place' imbued with a spirit of accessibility will be endowed with a multitude of experiences that will lead them to a memorable stage in their never-ending process of human fulfilment.

## 3. On the Challenges Facing Travelers with Functional Disabilities

At first glance, it may seem surprising that in today's world of almost limitless information technology possibilities a traveller in an electronic wheelchair, as in this case, is faced with the difficulties of organising the journey he or she has longed for. In recent years, a number of studies have been published on the development of various travel websites and the veracity of the respective information; see, for example, Jiang et al. [17]; Nod et al. [18]; Whitehead [19].

In this case, after many days of searching on booking.com, the service provider with the largest accommodation supply, one of the authors found the property (apartment) in the centre of Madrid that seemed to offer the best value for money, based on the site's search engine settings (although the disabled traveller was travelling to the destination with an assisting partner).

Here we must pause for a moment with the 'experience report' and reflect together on why a seasoned traveller, for whom technical accessibility and fundamental accessibility are both important, does not look for websites that list and make available a wide range of accommodations specifically designed to meet such needs, like, for example, www.handiscover.com/en-gb, accessed on 19 December 2022 [20].

The answer is infinitely simple: the wheelchair traveller, and later the traveller in this particular situation, is one of the authors of this study, a philosopher and practicing Buddhist on the one hand and a researcher in the science of travel on the other. It is for this dual identity that he particularly appreciates the work of Rupert Sheldrake and, in particular, his book named 'Unleashing the spirit of enquiry—10 inspiring questions for

science renewal' [21]. Inspired, among other things, by this, he decided to add to his list of desired experiences the experience of travelling on non-specialised tour operator trips, thus taking the risk of arriving at an apartment advertised as fully accessible and supported by photographs—and not being able to get into the accommodation he had paid for.

Of course, Sheldrake is not saying, either, that the researcher should risk his or her own, his or her partner's, or his or her peers' sanity indefinitely, but 'merely' demonstrates how much new knowledge and scientific facts can be gained, for example, by questioning the usual methodological frameworks and not treating them as substantive. It is worth noting that the thinking of the anarchist philosopher of science, Paul Feyerabend, who became disabled as a result of a plane crash, is also akin to this open-mindedness and open-endedness in the acquisition of knowledge [22].

It goes without saying then that, based on many years of experience in travel management, the respective author contacted the accommodation owner directly and all specific questions related to wheelchair transport and lifestyle were asked in detail in written English, and he received a reassuring answer. To be on the safe side, a telephone consultation was also made the day before the trip, during which it was discovered that the lady owner had difficulty with speaking English, contrary to the information on the website. Nevertheless, she confirmed that the property was fully accessible by electronic wheelchair and that the interior too was wheelchair accessible.

At this point, the 'Madrid story' must be interrupted once again to draw attention to some of the more important aspects of the accessibility situation being examined, which concern safety.

First of all, the statement may sound trivial, but the authors have to start by going around this particular hermeneutical circle [23,24], or rather hermeneutical sphere [25], and placing ourselves inside it, by stating that the existence of technical accessibility and fundamental accessibility for people who cannot do without *it* or *them* is a major security risk. Consider this specific situation: a wheelchair user with a functional disability (Figure 1) and his partner are faced with a flight of stairs of at least ten steps, which are not equipped with a device that can safely transport the electronic wheelchair, weighing nearly 120 kilos with its user, to the entrance of the apartment and back to the ground floor. The internal usability of the property itself is not even mentioned at this stage. The traveller is therefore faced with a situation that seems impossible to solve, which has arisen after the most careful travel planning 'manoeuvre', and which has replaced the long-planned week-long trip to Madrid with the horror of being without a roof. This sentence and the series of metaphors it contains may not fit into the accepted theoretical and empirical framework of an academic work, in this case a paper of academic value.

Here, the authors hope that the reader and the audience of experts will understand the depth of the layers of information described in the cases of 'enumeration' of the indispensable elements of the 'methodology' of philosophy as a way of understanding the world, as described by Hans Georg Gadamer [23], Martin Heidegger [10], and Karl Jaspers [24], among others. Paul Feyerabend goes even further than the classical philosophers cited, summarising, for example, in his iconic work *Against the Method*, that any scientific discovery or claim can never be exempted from the sources of explicit or implicit experience from the inner world of the person(s) formulating it [22].

One of the defining elements of Henri Bergson's philosophical legacy is the 'juxtaposition' of ontological and epistemological approaches—the title of Bergson's magnum opus, 'Creative Evolution' [25], itself attests to this highly open-minded approach, which the authors believe is akin to the fundamental holistic philosophical teachings of the Buddha [26]—and the development and presentation of a particular way of understanding existence. It is this: a person open to philosophical inquiry can choose between two basic ways of knowing an existent, existential thing. S/he can observe the 'object' from the outside, which is essentially what we call epistemology, or s/he can map it from the inside, which is, in a nutshell, the ontological approach. In other words, we can think in terms of philosophical 'models' for understanding the features of knowledge and its

relation to being [23]. Bergson, however, recognises the dangers of a shift of emphasis in diversification and therefore suggests that, after looking at the external features, we should not only try to draw theoretical conclusions about the relation of being and existing things, but actually try to 'get inside' the object under consideration itself. This type of 'sampling procedure' is, in the authors' view, similar to the basic structure of the analytic approach of the Madhyamaka 'philosophical school' based on the discourse on the proof of the emptiness nature of being and existence [27]. Briefly, this is achieved in the following way: for example, the discussants select an object at their fingertips, each of them perceives it according to his or her position in being, and on the basis of these perceptions describes the nature of the object of study on the basis of their personal feelings, experiences, and memories. Then, after having explored the various aspects, they try to reach a common position, taking into account that the relation to the object may, in principle, give rise to different interpretations and descriptions due to the multiplicity of physical and conscious positions. In other words, the aim of the debate is not the persuasion and/or defeat of the opponent that is perhaps common in the 'Western' culture, but the development of an ontological and epistemological explanation that is approximately acceptable to all, strictly in relation to the present moment [28].

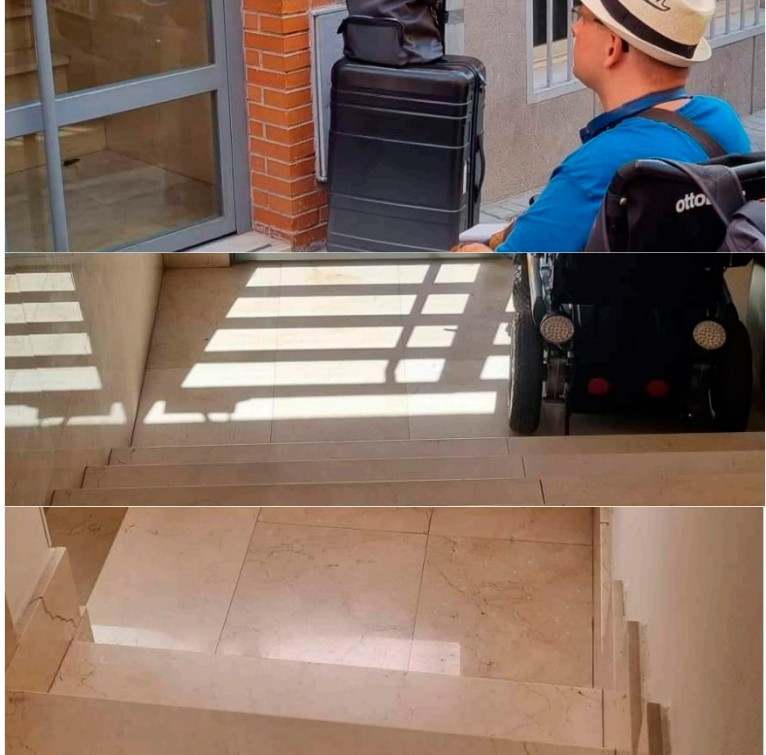

**Figure 1.** Virtual vs. physical reality. Source: the authors.

The authors of this study, also using the several-thousand-year-old practice of the briefly presented methodology, interpret, for example: the 'sets' of accessibility, safety, credible information, and travel. As we will see, this philosophical practice and theory mix is not alien to 'Western' philosophical scrutiny; suffice to mention in this place the approach worked out by Henri Bergson. It is true, however, that the application of meditation practices is not widespread in Europe and America, and it is nevertheless an indispensable part of Buddhist life philosophy practices [28].

Given that human consciousness, in the Bergsonian interpretation, is also 'in itself' the investigator and in a sense also the instrument under investigation, remembering, for example, the 'milieus' detected in the exploration of external features is an extremely important component of this type of cognition. The relationship between the external and

the internal world is interdependent, but our interpretation is that it maps our world along a non-hierarchical relation, similar to the Madhyamaka position already explained. This view is shared by Gilles Deleuze, an eminent scholar of Bergson's philosophy, for example, in his work titled The Bergsonian Philosophy [29]. In other words, the methodological dimensions of the science of travel can be complemented by the investigative approach of this philosophical methodology, since the traveller who seeks genuine experiences is curious, for example, about the external and internal experiences offered by the place and cultural milieu of his or her choice, as well as the experiences of novelty that organise these two in his or her consciousness.

In this way, it may be easier to understand the message of this study, which has a specific structure and message, based on real experiences, and the somewhat implicit nature of its real scientific values.

Coming back to the safety aspects, if the tenant cannot even get into the property itself, which has been found and 'walked around' and is said to be accessible, and may not have additional funds to rent another accommodation available to him in the same destination—indeed, in this particular case, the rent or the accommodation was refunded in full within a week by Booking.com, the seller of the accommodation—regardless of the type of disability (existential or functional), his or her personal security is simply at risk. Thus, it is clear that life can present a myriad of obstacles for travellers, and most of these obstacles are not even related to the existence of specific accessibility needs generated by functional disability.

Let us remember how the preceding two pandemic years have generated so many obstacles in the narrow dimensions of travel science alone and have made humanity aware of many more [30].

As researchers and thinkers, looking at the way the world is going, we have to conclude that we need to rethink and adjust our own perspectives in a deeper and more structured way, given the general picture of the world around us, of which the imagined nuance of relative security, which in the Western hemisphere was an indispensable part, now seems to be dissipated by the pandemic. This belief in security seems to have been ultimately shattered by the armed conflict in our neighbourhood, which is flooding our world, considered safe hitherto, with unprecedented obstacles. In other words, the understanding and sometimes necessary removal of barriers, or the design and creation of a built environment free of obstacles, is now clearly not only a civilizational achievement for the benefit of people with functional disabilities, at the 'mercy' of the majority society, but also a fundamental way of being human and an essential basis of man-generated activities.

By guiding ourselves back into the arbitrarily drawn coordinate system of fundamental accessibility and technical accessibility of travel, the authors are now confident that they have a vague idea of how much work lies ahead. If they want to apply it only to a narrow cross-section, or the beatific journey [31], this particular and intentionally eclectic endeavour of their previous case and research studies is of a philosophical nature [32,33]. However, they are only standing at the bottom of the stucco-covered staircase of the Madrid apartment, completely free of both fundamental and technical accessibility—more precisely, one of the writers of these lines is sitting there.

Travellers were faced with at least an equivalent obstacle when it was discovered that the owner of the property had very limited English language skills. Moreover, the lady owner was not willing to engage in what was in fact an unnecessary conversation via the mobile phone completely free of charge to her with the functionally disabled traveller or his partner. She asked to 'talk' about the accessibility of the apartment, which she still considered and stated to be accessible, via the WhatsApp application.

It is worth noting that the lack of communication accessibility itself has been one of the most pressing problems since the advent of organising travels in the online space [34], all the more so because the rise of mass travel has brought into tourism people from social strata who, for example, due to their age or the specificities of their national education system, have not acquired English or any other foreign language [33,35]. The scope and specific points

of investigation and framework of this study do not allow for a further detailed analysis of this issue, but the authors see the unmapped territory of the dimensions of communication accessibility and intend to extend the present paper in these research directions.

While the authors believe a brief mention of their empirical research on physical accessibility from a safety-focused standpoint along with the respective viability is important, they are aware that the inquiry into mobility-centred accessibility is only one 'continent' of the complexly understood 'worlds' of accessibility and usability. A much less explored 'continent' is that of ease of understanding, which inherently calls for providing people with mental disabilities, autism spectrum disorders, Down's syndrome, etc., with informative but also easy-to-understand readings on the content of a website or document [36].

Based on the interpretive domains of safety, ease of understanding, and accessibility (Figure 2) explored in the case study and lived experiences, the authors also conclude that one way to promote the elimination of language barriers, and elicit responses to these could be the increasing use of websites, information, signposting and warning signs with easy-to-understand graphics, photographs, pictograms, and simple text. Therefore, as we can see, the original—and unfortunately still not widespread—concept of easy accessibility holds much more potential for the holistic paradigm of accessibility than we might have originally thought.

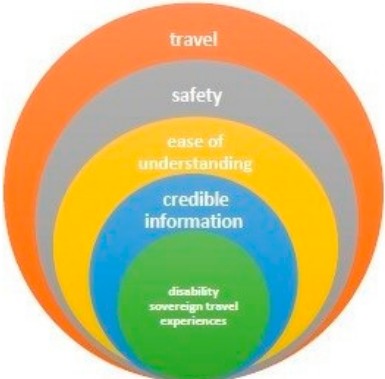

**Figure 2.** Domains of interpretation of safety, understandability, and accessibility. Source: edited by the authors.

Going back again to recalling and recounting the Madrid experience, communication with the owner in English did not become any clearer after installing the WhatsApp application. All that the travellers managed to understand was that the homeowner's brother would bring the keys and that they would use the non-wheelchair lift next to the stairs to get in. However, the owner made it clear that his brother was not able to speak English at all. After some two hours of waiting, a relative arrived, clearly illuminated by the smell and behaviour patterns, who, seeing the electronic wheelchair and the person in it, confirmed with eloquent Spanish and unmistakable gestures the fact, already obvious to travellers, that the accommodation was not, along these lines, of an accessible nature. To their credit, they assisted the mobility-impaired tenant in climbing the stairs to prove that at least the internal usability of the apartment was impeccable, and with the help of a local friend who had arrived in the meantime, while the travellers 'rested' for half an hour, they tried to find some way of getting the wheelchair up and down. This proved to be a mission impossible, just as the interior design of the accommodation was as far from accessible as the otherwise perfectly accessible Madrid milieu was from embodying a holistic approach to accessibility as the authors understood it.

To illustrate this, 'only' two important accessibility gaps are highlighted:

1.  The staff at Madrid airport specialising in helping passengers with special needs do not speak any language other than Spanish.

2.  The almost entirely barrier-free metro lines, built on a multi-level tunnel system that runs through the capital, do not boast a single sign in English to help travellers find their way. Thus, the lifts in the stations only announce the direction of travel and the place of arrival in Spanish.

It is not possible here to give readers a textual and detailed insight into how the author and his partner managed to fill the one-week trip to Madrid with more or less positive experiences, and for example how they managed to book a room in a truly accessible hotel located nearly 30 kilometres away from the centre of Madrid, access it, and commute from there to the tourist attractions in the centre of Madrid every day (Figure 3).

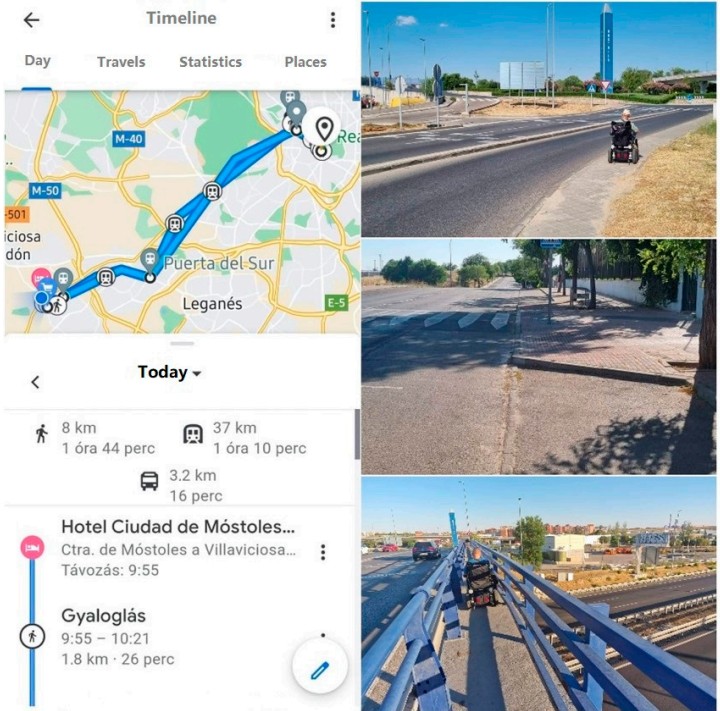

**Figure 3.** Desperate times, desperate measures. Source: the authors.

The authors are convinced that this highly original theoretical framework-mix will stand the test of scholarship, as well as their previous works, even if the aforementioned fertilizing lens of subjectivity in the philoscope is an original intellectual tool of inquiry that seeks to combine mainstream research methodologies and up to several thousand years of 'Eastern and Western' approaches to philosophy. The elements of reality 'under scrutiny' by the researcher–philosopher and, for example, the theories concerning them are placed in the philoscope as a lens, which, guided by the analytical-sensory aspects of human consciousness, is set in motion on the object of study. Consequently, a multidimensional set of images is unfolded by the resolving power of prisms of different colours and magnifications [16].

In this way, it may become clear to the reader why the authors openly adopt an eclectic, yet constructive and, they hope, empowering style and method of textual management, which is also not without precedent—just think of the 'methodology' of Socrates [37], cited by Plato's reminiscences, or of another great figure of the Jaspersian axis age [24], the founder of religion, Sakyamuni Buddha [9], who also often indulges in verbosity and even paradoxical questioning. In addition, the authors are introduced to the valuable, valid, and useful findings of the reduction ad absurdum approach of 2nd century Indian philosopher Nagarjuna, who followed in his footsteps [38].

In the next section of the paper, a brief empirical research analysis is presented, perhaps more in line with today's academic expectations than the chapter before, which hopefully

will be able to provide sufficient support for the personal, albeit universal, messages and thoughts of the theoretical corpus.

## 4. Empirical Support

The analysis of the empirical data used in the research took place with the consideration of comments made on accommodations on the booking.com website (the day of data collection was 13 October 2022). An obstacle to the research is the fact that the way booking.com operates and stores or deletes comments was out of the control of the authors, similarly to all websites of this kind that the authors had worked with during their previous researches. For this reason, a control test was also made, and it was found that several comments became unavailable the week after the data collection. Thus, the analysis is a hybrid one; it can mainly be taken as a qualitative analysis from which conclusions of a quantitative nature can be drawn. In this sense, the empirical analysis is part of the hermeneutical focus of the paper as well. With regard to this hybrid way of interpretation that the authors call philosophical [6], no academic methodology frequently seen in the scientific mainstream was used. The authors are aware of the fact that their paper is on the borderline of two sciences (philosophy and travel science) and special case studies. As it has already been mentioned, the new paradigm of fundamental/deep accessibility is definitely a subjective interpretation and enlargement of the concept of technical accessibility, and so the authors find it extremely important to articulate personal experiences, maybe less typically in academic life. This is in line with the international convention accepted by the UNO that guarantees the rights of people living with disabilities and is also meant to provide them with equal chances, still in effect today, as can be read from UN document in effect [39] (UN) and work by other researchers: "The study reveals that accessible tourism is of utmost importance for humans in today's world. In my opinion the biggest problem of accessible tourism, as regards services, is subjectivity, which means that companies are unable to fully satisfy the needs of all "impaired" persons at the same time, which makes the selection of the right target group important" [40].

For people with functional disabilities, the realisation of the received experience after the promised one (i.e., full accessibility not only at the level of promises but also in reality) is even more important than for people only with existential disabilities [6,7], because they have much less choice and it is much more difficult, time-consuming, and costly for them to find new accommodation in a given location if the one they originally booked does not meet their expectations and accessibility requirements. This drastically reduces the quality of their experience (and even the amount of time they possess to spend experiencing it), which can make travelling not only a beatific experience [31,41] but also a source of frustration. In other words, safety (of their money spent, and of their expected experiences, and not least, as a consequence of the events mentioned in the theoretical section, even of their physical integrity) is perhaps even more important for travellers with functional disabilities [42].

In order to avoid inconveniences and guarantee safety, travellers with functional disabilities can use the services of organisations that can guarantee the existence of real accessibility. At a European level, perhaps the most important such organisation is ENAT, European Network for Accessible Tourism [43]. ENAT is a non-for-profit association of tourism businesses, organisations, individuals, and the public and civil sectors, registered in 2008 (after some two years of preparatory work) with the explicit aim of identifying and evaluating best practices in accessible tourism, making all tourist destinations as accessible as possible, and promoting the creation and promotion of accessible tourism services and products (originally in Europe, now worldwide) [44]. Launched in 2006 as a project-level initiative, ENAT initially involved 9 founding partner members from 6 European countries, including VisitBritain—the UK's tourism body—and ONCE, the foundation of the Spanish National Organisation for the Visually Impaired. The total number of members at the end of 2020 was 222 from 58 countries (four of them in Hungary).

Of particular importance for Madrid is the fact that it was included in the ENAT's list of European Accessible Cities several years ago, in 2017, so it is a legitimate expectation that

accommodation labelled as accessible in Madrid should meet the expectations of travellers with functional disabilities [45].

On booking.com, the world's largest accommodation website, there were around thirty hits for accessible accommodation in Madrid other than hotels at the beginning of October 2022. One property's own description has been on the site since April 2002 (this is probably a typo and April 2020 is the correct date, if only because it has received just over a thousand reviews), and the shortest registered property has been available on booking.com since February 2021. On average, the hotels in question have been listed on booking.com since autumn 2013, so one would expect them to be (also) professional in terms of accessibility.

The platform booking.com provides information on the following accessibility factors:

- raised toilet seat;
- lower adjustable bathroom sink and mirror;
- visual aid: Braille;
- visual aids: tactile signs to help you get around;
- voice control assistance;
- an emergency alarm in the bathroom at the right height;
- toilet with handrail;
- wheelchair accessibility;
- the whole facility is wheelchair accessible;
- the facility is located entirely on the ground floor;
- the upper level is accessible by lift;
- shower chair;
- roll-in, wheelchair accessible shower.

It is thought-provoking that although the search for accommodation was filtered by 'accessible facilities', there was one accommodation in the survey that did not have accessibility listed as a property feature, and another that did not have accessibility listed on the main page, but the sub-headings included a handrail in the toilet, access to the upper floor by lift and wheelchair accessibility, and six places only stated that the accommodation was accessible, without any specific description. In one place, a discrepancy was found in the description of the accommodation: both the accessibility of the whole facility and the fact that the upper level can only be accessed by stairs were among the pieces of information shared. Note that not all the categories listed by booking.com were mentioned, and none of the accommodations offered (or they did not consider it important enough to mention that they offered) shower chairs, wheelchair accessible showers, voice, or visual assistance (Figure 4).

Not one accommodation was found that included a link to the website of an independent certification body whose profile is to assess the accessibility of facilities (e.g., the internationally known Handiscover, or Access4You, which is becoming increasingly well-known in Hungary and the European Union) [46,47] or even the ENAT website [43].

All of the selected hotels uploaded image(s) to their websites, typically 8–9 images on the main page, and additional images (35–40 on average) were available with a single click. Based on these images, the authors wanted to determine whether the accommodation was actually accessible or only accessible on the level of promises. When examining the images, the criterion was how many of them clearly showed accessibility (wide enough corridors, wide door openings, fixed and folding handrails in the toilets, etc.). It was found that there were no accommodation facilities where the pictures could be used to determine with a high degree of certainty that the accommodation was actually accessible, but there were a good number of pictures that at least raised doubts, and in addition, in none of the pictures on the main page was it possible to infer accessibility at any level, as the information on this could only be found by clicking on the additional pictures.

A further problem with the images was that many of the dozens of photographs often included the same images more than once, or the same room with minimally different settings, so that the reader gets redundant images with virtually the same information, while no single image reveals the existence of accessibility.

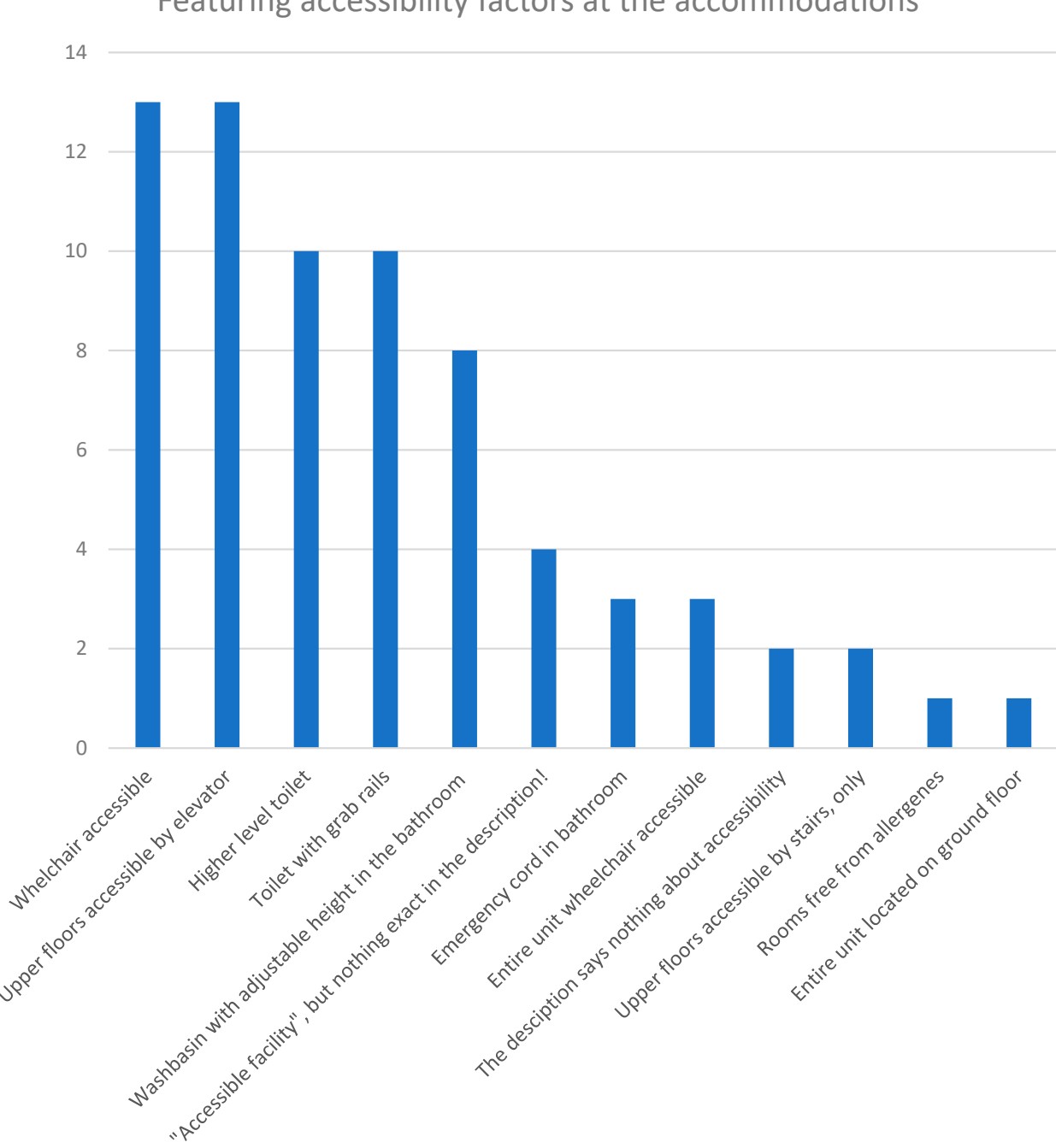

**Figure 4.** Accessibility factors on the websites of the surveyed accommodation facilities. Source: www.booking.com (accessed on 18 January 2023).

Language accessibility and the ability of staff to communicate in English can also be indicated in the description of the accommodation on booking.com. Interestingly, there were two hotels where this was self-reported: it may be simple inattention, but where (in a third case) Spanish and Farsi are listed as the languages spoken by staff, it is unlikely that the lack of English communication skills was due to forgetfulness (English was not included as an accessibility criterion because of the authors' language skills, but for the simple reason that it is required information about the accommodation on booking.com).

The survey also analysed guest records related to the accommodation. In total, 26 accommodation establishments in Madrid declared as accessible received no less than 45,000 reviews, ranging from a minimum of 115 to a maximum of 7473. It was not possible

to read this many reviews in detail, and all reviews were searched by keyword for the following terms: accessibility, accessible, wheelchair, handicapped, disabled, disability, disabilities, impaired, blind, deaf, autist, disease, syndrome, malfunction, disorder, language problems, communication, didn't understand.

The authors searched for comments written in all languages, and where the language was not English, booking.com's own translator was used [48]. As some words may have a meaning that is not accessibility-related, only those words were searched that had the appropriate meaning, for example, the mention of 'good access' to an accommodation in a good location and easily accessible by public transport was ignored, as was 'blind' if it referred to a window blind and not a blind person.

There were not many comments on the accessibility—and independent usability—of the accommodations by travellers with functional disabilities (unfortunately not one positive; where there was a mention, it was always to point out the shortcomings), but more on the usability of the interior (Figure 5). It must be remarked that the number of hits is probably reduced by the fact that only words spelled correctly were considered, while misspelled and alternative versions were not (e.g., ilness instead of illness, comunication instead of communication, or short sighted and shortsighted instead of short-sighted, etc.). In each case, a positive test was carried out with words whose occurrence was certain, to check that the system was working correctly. The authors deliberately did not use the categories provided by booking.com because they considered it unlikely that standard professional terms in a comment would be used. The names of accommodations were deliberately not mentioned due to data protection considerations.

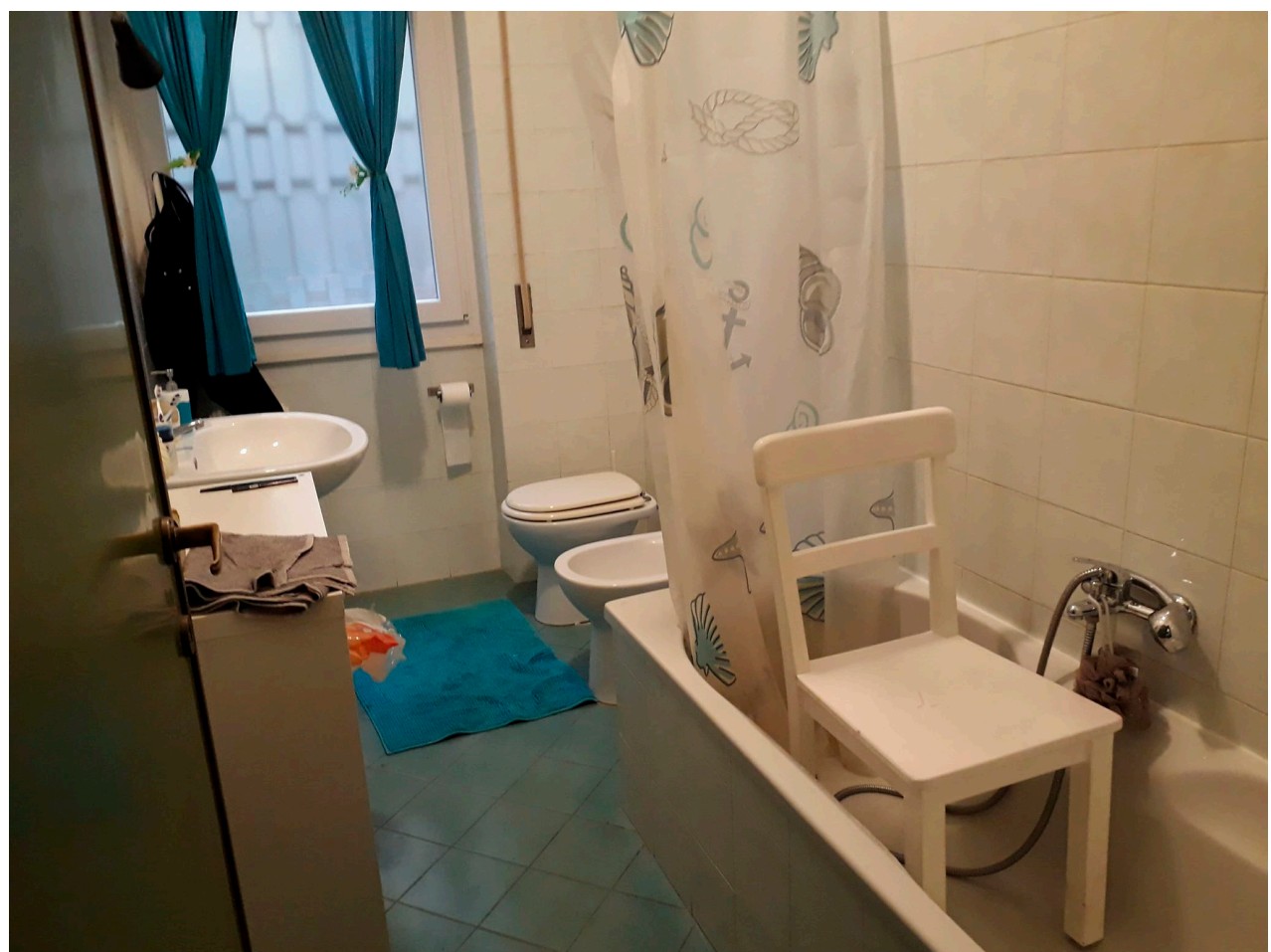

**Figure 5.** 'Making' a bathtub accessible by travellers in a hotel advertised as 'accessible', Rome, 2020. Source: the authors.

Some typical problems:

- "The positioning of the toilet is unfortunate, a tall person can't fit."
- "The only problem was the size of the bathroom door, but we had expected that."
- "The bed is difficult or impossible for disabled people to use on their own."
- "The bed is upstairs, where you can only go up the stairs."
- "The bathroom was too narrow." (This was a problem mentioned in several places, with two reviewers specifically mentioning the impossibility of wheelchair access in one accommodation).

Communication difficulties, which have a serious impact on travellers' sense of security, were also mentioned in a few occasions, with some French guests complaining about the lack of French language skills (although English was spoken at the accommodation), some staff not speaking English being a problem, and some complaining about the lack of Spanish spoken by staff at the accommodation (interestingly, the lack of Spanish language skills was mentioned in two places).

The evaluations follow the natural subjectivity factors, which are also formulated in the theoretical section [49], but the authors are convinced that these do not affect the scientific evaluability of the paper or this short (case) study at all. There have been reports of both fantastic and tragic language skills in the same accommodation establishment. This may be a problem of the language skills of the staff working a particular shift, but the natural thing to do in the accommodation services sector of a large European city would be for everyone to be able to communicate in English. Equally, the width of the transport corridors within the establishment may be acceptable to one user and unacceptable to another, which is why it would be helpful if the photographs clearly showed whether the accommodation was truly accessible. Negative ratings for communication and language problems may also be due to the fact that the assessor came into contact with the only person in the accommodation who was unable to communicate with him/her (even if s/he was there only temporarily, as a substitute), and normally has no problem in communicating with guests.

Since the upload of data is by self-report, there is a high risk of error or even deliberate embellishment of the facts; if boooking.com insisted on certification by an independent rating body, at least in the area of accessibility, which seems to be a particular sticking point, perhaps people in need could choose accessible accommodation with greater confidence.

Based on the results, and drawing conclusions from the ratings, accessibility does not seem to be a very important aspect of travellers' choice of accommodation, as even for accommodations that are explicitly advertised as accessible, a small percentage of comments criticised, for example, the difficulty of getting around in a wheelchair.

Of course, there are serious limitations to this investigation:

- the subjectivity of respondents: it is possible that not everyone's threshold for a particular problem is the same, or even if this threshold is reached, they will react differently;
- a small sample was used, and the authors plan to extend the research to other destinations and even to other types of accommodation and other accommodation websites (szallas.hu, trivago.com, etc.);
- the list of keywords searched for in comments can and should be extended;
- it was not clear from the responses whether the evaluations were actually given by people with disabilities or just by people who are sensitive to the problem and want to draw attention to the shortcomings.

Consequently, the further enlargement and deepening of the research is indispensable, by, e.g., an attitude survey carried out in the future [50].

The authors are aware of the limitations posed by the character of a review paper, which has inspired them to launch, just on the basis of the partially empirical experiences gained here, a large-scale research with an extended international database involving the most visited European destinations which, in all probabilities, will scientifically and quantifiably support the statements in this paper.

## 5. Conclusions

As is perhaps customary for the authors, this essay of theirs does not follow the usual linear structure, but adopts a seemingly eclectic but primarily medial structure of philosophical inquiry [7]. Furthermore, the stream of thought, which can be partly defined as a case study, is also a methodological experiment, different to some extent from mainstream research, which focuses on outlining a somewhat subjective experience of living and a holistic need for security and seeks to present both philosophical and empirical support for this. For their part, the authors do not wish to implement the coherence of these two fields of inquiry in a textual-didactic-manner, given that it clearly concludes from the application of Buddhist hermeneutic inquiry [4,16]. That is, since the authors are aware of the professional skills and needs of the academic readership, it is left to them to integrate the findings and results of the two modes of inquiry in their own way.

The paradigms of fundamental accessibility and technical accessibility are organised in a practical framework, which means that an attempt was made to bridge the dimensions of theory to the realms of reality. In the world of science, the use of subjective experience as a raw material has very limited possibilities, or one might say, is not a common practice. However, the characteristics of case studies may allow some leeway for the use and publication of individual experiences and lessons. The authors are convinced that for travelling, being a traveller in itself implies accessibility and, paradoxically, it serves the realisation and knowledge of accessibility and also entails the demand for fundamental accessibility. Existential disability as a natural state of being can be linked to this without any particular thought effort, since it is argued that the relation of being and existence of humans can be understood as a continuous journey on the one hand, and, on the other, that the realisation of this activity can be understood and described as a continuous activity of accessibility through the understanding and living of existence.

The set of Madrid experiences, as recounted by one of the authors (and his companion), is a 'boatload' of lessons, providing both applied philosophy and rigorous empirical research with a wealth of thought-provoking and measurable data and information.

However short human memory may be, what is perhaps a lasting and memorable experience for all of us is how the COVID-19 pandemic tore open one of the veils covering the layer of reality that kept the happier half of humanity in a false sense of security. We have discovered that a microscopic organism, incomprehensible to us, has, in order to survive, 'used' the multitude of travel and accessibility machines constructed by humans today to travel around the world in a matter of moments and, in fact, has closed the 'transport corridors' of our entire environment to us in a way that is more effective than the dangers which our senses can detect. In this way it has generated social and economic crises still affecting our lives today, despite the fact that these areas of our existence were thought to be also fully accessible, and so less filled with dangers [51]. Furthermore, the COVID-19 pandemic has brought to the surface a plethora of lessons and things to be analysed, concerning, among other things, the need to make the safety focus of our future plans more sophisticated [52,53].

In theory, the Madrid infrastructure, which is indeed physically perfectly accessible, should have guaranteed the traveller with accessibility needs—and functional disabilities—and his partner a visible and perceptible natural way of the security of existence (examined from several aspects) and, consequently, the beatific nature of having and enjoying undifferentiated experiences.

The authors are convinced that idealism is the furthest thing from their minds, because, as can be seen from the meeting points of the parts of the paper, the theoretical and empirical statements not only complement each other but also organise the two 'existence-analysing' approaches. Moreover, it is close to the philosophical and scientific worlds of meaning by Pierre Hadot [54] and Paul Feyerabend [22], i.e., it is a plastic representation of the often-questioned legitimacy of transdisciplinary approaches.

With all this said, the authors are aware of the limitations, or should they say obstacles, of their experiential research. The data content of the section on the accessibility aspect of

websites—and consumer comments on the presence or absence of accessibility—may be minimal, but the amount of information available has simply made the paper functionally handicapped. This, however, has given authors an appetite for a real 'deep dive' into the third sphere of existence of the Internet [55] and, at the same time, has given them the opportunity to develop the theoretical philosophical part in a much more structured way.

Finally, attention must be drawn to some key practical moments that highlight, among other things, the shared social and individual responsibilities that, in this case, hinder rather than facilitate the development of a more accessible and inclusive world:

1. The lack of control of accommodations advertised as accessible, or the need for it, for example in the case of the booking.com portal examined in the paper, is essential.
2. Both the United Nations and the European Union are categorically in favour of consumer rights and are trying to make it legally binding for member states to guarantee these rights. However, the existence or lack of an accessible environment is, as far as the authors are aware, a basic requirement only at the level of documents containing international recommendations, and there is almost no practical monitoring of these.
3. The integration of the experience of the disabled people concerned, either self-taught or through the comments and concrete proposals made by their professional organisations and associations, into the development of concrete legal and accessibility 'standards' is an essential precondition for the development of these codes.

In light of this, the authors see explicit support for the applicability of the personal field of experience in their study, and for the seemingly dialectical but non-hierarchical juxtaposition of philosophy and scientific inquiry.

As a final thought, it should be noted that, following the experience gained during the preparation of this review paper, the authors have entered the implementation phase of a 'classical' study, one of the central points of which will be a quantitative, large-scale data analysis.

**Author Contributions:** Conceptualization, J.F. and Z.R.; methodology, J.F. and Z.R.; software, J.F.; validation, J.F., Z.R. and L.D.D.; formal analysis, J.F. and Z.R.; investigation, J.F. and Z.R.; resources, L.D.D.; Z.R.; writing—original draft preparation, J.F., Z.R., J.K., Z.F.-F. and L.D.D.; writing—review and editing, J.F. and Z.R.; visualization, J.F. and Z.F.-F.; supervision, Z.R. and L.D.D.; project administration, Z.R.; funding acquisition, not relevant. All authors have read and agreed to the published version of the manuscript.

**Funding:** This research received no external funding.

**Institutional Review Board Statement:** Not acceptable.

**Informed Consent Statement:** Informed consent was obtained from all subjects involved in the study.

**Data Availability Statement:** The study did not report any data apart from booking.com's website, where the (dynamically changing) number of reviews were mentioned.

**Conflicts of Interest:** The authors declare no conflict of interest.

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
