# Peer review of "The Dialectics of (Deep) Accessible Tourism and Reality—Hermeneutics of a Journey to Madrid"

_sustainability, doi:10.3390/su15043257_

Round 1

Reviewer 1 Report (New Reviewer)

Dear authors,

Although your paper does not follow the parameters of positivist research, it makes, however, a contribution by offering reflections based on personal experience and questioning the way "accessible accommodation" is advertised by accommodation booking platforms and the contradiction between erroneous discourse and reality. 

In my view, the paper can still be improved. I'll make some suggestions:

- suggestion for the title: The Dialectics of Accessible Tourism and Reality – Hermeneutics of a Journey to Madrid. The word "dialectics" points to the contradiction and, on the other hand, is more in line with the philosophical viewpoint assumed in the paper. Accessible Tourism is a stronger topic in the context of the paper than "Accessibility". Strong topics are important to get included in searches on the Internet and, therefore, increase the probability of the paper for being read and cited.

- Some caution should be taken in the use of language, especially in what concerns the use/non-use of inclusive language.  In the paper, you use 11 times the word "man" and several times the pronoun "him". Just as an example, consider the sentence «... “philosophy” has remained an experience of man conflicting him with events, which dignifies him as man...». Maybe it could be rewritten as «... “philosophy” has remained an experience of humans conflicting them with events, which dignifies them as humans...».

Another example: «the evolutionary history of man naturally includes the potential for adaptation, socialisation, mutual assistance and, not least, the transformation of the living environment, i.e. it is one of the intellectual drivers that make man a man». Suggestion «the evolutionary history of humans naturally includes the potential for adaptation, socialisation, mutual assistance and, not least, the transformation of the living environment, i.e. it is one of the intellectual drivers that make humans what they are».

Another example: «He can observe the “object” from the out-195 side, which is essentially what we call epistemology, or he can map it from the inside...». Suggestion: «He/she can observe the “object” from the out-195 side, which is essentially what we call epistemology, or he/she can map it from the inside...».

- There is an apparent contradiction in the narrative: lines 271-274 «Travellers were faced with at least an equivalent obstacle when it was discovered that the owner of the property had very limited English language skills. Moreover, she was not willing to engage in what was in fact an unnecessary conversation via the mobile 273 phone...».  In this case the owner of the property was a "she". Later (lines 308-309) you write «However, the owner made it clear that his brother was not at all able to speak English». The pronoun "his" points to a masculine owner. 

Probable typos: line 303: «Domains of interpretation of safety, accessibility and accessibility». 

Line 448 «Nort one accommodation was found...».

Improvement of the conclusions:

- For the first time you speak about Covid-19 in the conclusions. I would not add a new topic to the conclusions.

- The paper would benefit from the inclusion of "Practical Implications" in the Conclusions, pointing to the contradiction of the discourse of the commercial agents and the reality. This is something that needs to be addressed by tourism destinations and tourism industry. Here you can again state that the "accessibility factors" announced by the accommodation booking platform Booking.com need effectively to be assured by the accommodations that advertise themselves on the platform as "accessible". International Organizations/Institutions such as the United Nations and the European Union pledge for a more accessible and inclusive world and it is time to transform the discourse into measurable actions. 

Author Response

Dear Reviewer,

first of all, let us thank you for your effort, the time spent on reading our paper and the recommendations for improvement. Below please find our reactions, replies to the issues raised.

We thoroughly revised the paper, applying the track changes function for an easier detection of the changes, and we also refer to exact locations in the text, by lines, where applicable.

As regards the suggestion for a new title, we accepted your version and the paper is now entitled “The Dialectics of (Deep) Accessible Tourism and Reality – Hermeneutics of a Journey to Madrid”.

You mentioned that “some caution should be taken in the use of language, especially in what concerns the use/non-use of inclusive language”.  In the paper, the use of the word "man" has been replaced by human or humans, to have a more inclusive text. Also, “he” as a general personal pronoun has been changed to s/he, where appropriate.

As regards the contradiction in the narrative: lines 271-274 in the original version, “Travellers were faced with at least an equivalent obstacle when it was discovered that the owner of the property had very limited English language skills. Moreover, she was not willing to engage in what was in fact an unnecessary conversation via the mobile phone...”, the text has been re-written to make it more understandable. 

The two probable typos: “Domains of interpretation of safety, accessibility and accessibility” and “Nort one accommodation was found...” were typos indeed and have been corrected into “Domains of interpretation of safety, understandability and accessibility” and “Not one accommodation was found...”, respectively.

Improvement of the conclusions: so as this part should not be the first time the authors speak about Covid-19, we have modified the text, implemented and explained the philosophical and scientific necessity of remembering and recalling the COVID-19 “problem” through a more extensive explanation of the specifics of Buddhist hermeneutics (a new paragraph containing this issue, among other things, can be read from line 99 to line 122 in the revised version of the paper).

Concerning your recommendation that the paper would benefit from the inclusion of "Practical Implications" in the Conclusions chapter, pointing to the contradiction of the discourse of the commercial agents and the reality, the respective chapter is now finished with these, from line 672 to line 692 in the revised version of the paper.

The style, the English language of the paper has been refined, some misspellings corrected, using the services of an external proof-reader.

Thanking for your cooperation once again, on behalf of all authors,

Yours sincerely

Zoltán Raffay

Correspondence author

17 January 2023

Reviewer 2 Report (New Reviewer)

This article does not meet the scientific criteria (methods, discourse, literature review) for publication in Sustainability.

Reviewer 3 Report (New Reviewer)

This is a difficult paper to critique.  While the topic is important and there may be very important findings, it is very hard to read through it, since it is written in a style that is a bit unnecessarily philosophical and flowery.

A clearer writing style and clearer discussion of methods and findings would be much more interesting to readers. As it now stands, I fear that few would want to read the article, since the flowery and philosophical style of writing overwhelms the methods used and the importance of the findings.

It should be rewritten in a way that is understandable to readers and enticing, making readers want to read the article.  As it now stands, it is unpleasant to read, even if the data and findings are of value.

Author Response

Dear Reviewer,

first of all, let us thank you for your effort, the time spent on reading our paper and the recommendations for improvement. Below please find our reactions, replies to the issues raised.

We thoroughly revised the paper, applying the track changes function for an easier detection of the changes, and we also refer to exact locations in the text, by lines, where applicable.

We accept your remark that this is a difficult paper to read, and the fact that this is partly due to the style that was a bit unnecessarily philosophical and flowery. We have tried to “fine-tune” the text to make it easier to understand. For example, we have given our paper a more striking and attention-grabbing title than the original, the paper is now called “The Dialectics of (Deep) Accessible Tourism and Reality – Hermeneutics of a Journey to Madrid”. We would, however, like to note that the essentially philosophical-hermeneutical nature of the wording has not, to the best of our knowledge, caused any particular problems in our previous publications (including those of the MDPI family).

The style, the English language of the paper has been refined, some misspellings corrected, using the services of an external proof-reader.

Thanking for your cooperation once again, on behalf of all authors,

Yours sincerely

Zoltán Raffay

Correspondence author

17 January 2023

Reviewer 4 Report (New Reviewer)

While the manuscript shows the authors´ deep knowledge of philosophy, hermeneutics, and other disciplines and represents a bold line of inquiry rooted in relevant field research, I am afraid it in its current form does not meet the requirements of an academic article. Major revisions are recommended. The authors are advised to consider the following issues:

1.) It seems unclear what the actual research question that the article is addressing is. This should be explicitly specified in both the abstract and the introduction.

2.) Since Sustainability is a scholarly journal, it is not quite clear what the authors mean when they state that their intention with this article is to present their research to "the public" (line 52). Do they they mean the "academic public?" Or, the general public? If the latter should be the case, I have doubts that this is what academic articles should strive for.

3.) The authors seem to repeatedly juxtapose the philosophical and the academic (for example, lines 46 and 55). However, I do not think these concepts necessarily are opposites (or, incommensurable categories that would need to be contrasted).

4.) The Buddhist hermeneutical approach that the authors repeatedly refer to is never explicated. Though there is a reference to the Madhyamaka school of thought, it is rather brief (lines 204-215) and if to be used as the methodology for the article, it nees further elaboration. 

5.) The phrase "superficial picture" (line 113) is not the most adequate self-description of one's work in an academic journal.

6.) It would be helpful for the reader if the authors add a section in which they would situate themselves and specify the perspective and context they are writing from. In particular, what is the added value of having 5 authors for this article? Does each one of them bring in something unique?

7.) The manuscript should be revised for coherence and structure as the authors are often jumping from one topic to another. Their reference to what they call a "medial structure" (line 559) of the article can be no legitimate reason for incoherence.

8.) The authors never bring together the philosophical and the empirical support for their case (see lines 563-64). It looks like two independent lines of argument are being followed in the article.

9.) Case study is a very particular genre which follows specific rules. Also, here are various types of case studies. Furthermore, case studies have both their pros and cons. The article should reflect at least briefly on these aspects.

Author Response

Dear Reviewer,

first of all, let us thank you for your effort, the time spent on reading our paper and the recommendations for improvement. Below please find our reactions, replies to the issues raised, following the numbering of the problems.

1.) It seems unclear what the actual research question that the article is addressing is. This should be explicitly specified in both the abstract and the introduction.

We referred to this problem in the Abstract and the Introduction chapters:

As our research is a specific case study based on the participatory methodology of disability studies (involving the people concerned) and the non-linear, thought-provoking and awareness-raising practices of Buddhist hermeneutics, following the schemata of a philosophical, qualitative “research methodology”, our hypotheses are explained and justified in detail in the introductory section.

As our research is a specific case study based on the participatory methodology of disability studies and the non-linear, thought-provoking and awareness-raising practices of Buddhist hermeneutics, it follows the schemata of a philosophical, qualitative “research methodology”, whose initial hypotheses are as follows:

the owners and operators of the world's leading accommodation distribution site, booking.com (and TripAdvisor, used as a control), are less aware of the relationship between accessibility and accessibility. That is, the real accessibility needs of disabled travellers and their companions. Our second hypothesis is that an internationally accepted and mandatory “reality check” for advertised and defined accessible accommodation has not yet been developed by the rule makers towards accommodation providers, although the need for this would be essential based on our quantitative research.

2.) Since Sustainability is a scholarly journal, it is not quite clear what the authors mean when they state that their intention with this article is to present their research to "the public" (line 52). Do they they mean the "academic public?" Or, the general public? If the latter should be the case, I have doubts that this is what academic articles should strive for.

We have made it clear that we address this publication to the academic sphere:

“First, the authors emphasise that the academic readership should note: their paper is an investigation with a special hermeneutic-dialectical focus…” (lines 51-52 in the revised version of the paper); “That is, since the authors are aware of the professional skills and needs of the academic readership, it is left to them to integrate the findings and results of the two modes of inquiry in their own way.” (lines 625-627 in the revised version of the paper).

3.) The authors seem to repeatedly juxtapose the philosophical and the academic (for example, lines 46 and 55). However, I do not think these concepts necessarily are opposites (or, incommensurable categories that would need to be contrasted).

The footnote inserted in line 75 of the revised version is an explanation to this: “To avoid misunderstandings, let us be specific: this is not to contrast the two approaches, but rather to illustrate the complementary non-hierarchical relationship between the two aspects of investigation in this context.”

4.) The Buddhist hermeneutical approach that the authors repeatedly refer to is never explicated. Though there is a reference to the Madhyamaka school of thought, it is rather brief (lines 204-215) and if to be used as the methodology for the article, it nees further elaboration.

Accepting your critical comment, the above modification has been added to the introductory part of the paper, from line 99 to line 116 in the revised version:

The main characteristics of Buddhist hermeneutics, which the authors also use as a quasi-research method, are, without claiming completeness, the following. In its examination, definition and understanding of a given "existing entity" and its contexts, it explicitly avoids the linear and hierarchical approach, as well as the dualistic approach still used in mainstream positivist methodologies [4]. It strives to encourage readers, investigators and thinkers to continuously think and to develop a wide range of reflexions by “jumping” between different sub-disciplines. It is important to stress that Buddhist hermeneutics is both theoretical and practical. However, it explicitly avoids single-person or individual “investigations”, knowing that human consciousness relate to and interpret reality in different ways. As a consequence, the aim is primarily to bring the “contexts of reality” of the persons under investigation closer to each other. Last but not least, it seeks to maintain and demonstrate the positive aspects of alert consciousness not only in theory but also in practice. This way, “meeting” the requirements of the original teaching of the Buddha, it also aims at the improvement of the ability to remember. As an illustration of this there will be, for example, a brief reflection made in the case study summary to the shock to tourism induced by the COVID-19 pandemic and the need to remember the given trauma both on the individual and community level. All this is briefly discussed in the conclusion chapter of the paper, thereby also pointing to the relevance of the aforementioned seemingly eclectic text corpus construction mode, which is both an indispensable component and practice of Buddhist hermeneutics.

5.) The phrase "superficial picture" (line 113) is not the most adequate self-description of one's work in an academic journal.

Accepting your critical remark, e above modification has been added to the introductory part of the paper: “Thus, what the authors’ community calls the incomplete, mosaic draft picture is intended to refer to the juxtaposition of different interpretative surfaces.” (Lines 118-119 in the revised version.)

6.) It would be helpful for the reader if the authors add a section in which they would situate themselves and specify the perspective and context they are writing from. In particular, what is the added value of having 5 authors for this article? Does each one of them bring in something unique?

Accepting your critical comment with the intention of correction, we have answered the relevant question you asked as an addition to our “introduction” to Buddhist hermeneutics (in paragraph 5 of the Introduction chapter, lines 99–122 in the revised version).

7.) The manuscript should be revised for coherence and structure as the authors are often jumping from one topic to another. Their reference to what they call a "medial structure" (line 559) of the article can be no legitimate reason for incoherence.

Following your constructive critical suggestions and comments, we have significantly modified and supplemented the abstract and the introduction. We also provide a full explanation of the use of the medial structure in a more detailed description of Buddhist hermeneutics as a “legitimate” research methodological configuration.

8.) The authors never bring together the philosophical and the empirical support for their case (see lines 563-64). It looks like two independent lines of argument are being followed in the article.

For our part, we do not intend to textually - didactically - achieve coherence between these two fields of inquiry, given that the application of Buddhist hermeneutic inquiry clearly follows from. That is, being aware of the professional skills and needs of the academic readership, we leave it to them to decide how to integrate the findings and implications of the two modes of inquiry. (Lines 623–627)

9.) Case study is a very particular genre which follows specific rules. Also, here are various types of case studies. Furthermore, case studies have both their pros and cons. The article should reflect at least briefly on these aspects.

We are aware that case studies can be classified into different types, we have chosen and developed a very specific 'mix', and in this there are two components, one is the participatory approach of disability studies, in which the lived experiences of existence and life of the person(s) concerned are also expressed in a specific, one might say subjective way, and all these also reflect aspects of interpretation that are significantly informed by disability. The second component, closely related to this, is the practical application of Buddhist hermeneutics, whereby the 'practitioner' inevitably weaves his or her own experiences into the picture of the interpretative horizon of the subject in question, and then the findings of these experiences are interpreted in a dialectical, specific discussion in a community. In this case, the aim of the debate is not to “defeat” one of the participants, but to find a common position on the interpretation of the dimension in question.

The style, the English language of the paper has been refined, some misspellings corrected, using the services of an external proof-reader.

Thanking for your cooperation once again, on behalf of all authors,

Yours sincerely

Zoltán Raffay

Correspondence author

17 January 2023

Reviewer 5 Report (New Reviewer)

the research is fine and the details about the topic is decent, The experiences gained and analyses made by the authors 31 make it clear that technical accessibility. the work has been reviewed appropriately for publishing in sustainability. 

Author Response

Dear Reviewer,

first of all, let us thank you for your effort, the time spent on reading our paper and the recommendations for improvement.

Thank you for your positive comments and support. In particular, the acceptance of the philosophical and scientific methodological “mix” we have adopted is still not self-evident.

What we have changed is that the style, the English language of the paper has been refined, some misspellings corrected, using the services of an external proof-reader.

Thanking for your cooperation once again, on behalf of all authors,

Yours sincerely

Zoltán Raffay

Correspondence author

17 January 2023

Round 2

Reviewer 4 Report (New Reviewer)

Thank you for considering my comments! I think you have properly addressed the issues I brought up in my review. The only concern I still have is the lack of interconnection between the "philosophical" and the "empirical" approaches in the article. To put it simply, the text reads like two separate articles. As I see from your response to my comment no. 8, you are aware of this issue but seem to view it as an asset of their article. I as the reader think otherwise.

In any case, I believe the revisions have helped to improve the text.

This manuscript is a resubmission of an earlier submission. The following is a list of the peer review reports and author responses from that submission.

Round 1

Reviewer 1 Report

Dear authors, I have reviewed the entire paper, I think that there is no substantial changes in both content, structure, and updated studies.

This paper adds nothing new to literature review. Only authors added 15 references, in fact, some of them are very old. I did not understand it when I told you add updated authors. We are in 2023, please.For instance: 

Kropotkin, P.A. (1902). 

Adler, A. Életismeret (Understanding Life) Kossuth Könyvkiadó, Budapest, 1998 608 

Gehlen, A. Az ember – természete és helye a világban (Man. His Nature and Place in the World). Gondolat Könyvkiadó, Budapest, 609 1976 

Feyerabend, P. K. Three Dialogues on Knowledge. Oxford-Cambridge, 1991, Blackwell, p. 146. (https://ar-588 chive.org/stream/PaulFeyerabend/Paul+K.+Feyerabend+Three+Dialogues+on+Knowledge_djvu.txt) 

Darcy, S.; Dickson, T. J. A Whole-of-Life Approach to Tourism: The Case for Accessible Tourism Experiences, Journal of Hos-592 pitality and Tourism Management, Volume 16, Issue 1, 2009, pp. 32-44 593 

Richard Nance, R. Buddhist Hermeneutics. DOI: 10.1093/obo/9780195393521-0155 594 

Gadamer, H-G. A filozófia kezdete (The Beginning of Philsophy). Osiris, Budapest, 2000 

Reviewer 2 Report

YI think the authors reflected my point of the review understandably.   But, the authors should justify the results, analyzing and comparing the previous study. For example, which parts of the study are consistent with the results of the previous study or not.

Reviewer 3 Report

Dear Authors,

Unfortunately, I’m not able to recommend the revised version of the manuscript for publishing. Travel and Tourism are data driven industries. Even if ignoring the mixture of the concept’s distinct layers (General world philosophy; Personal Travel diary; Analytical effort on the issue of online accommodation listings’ information on accessibility), the manuscript is still unbalanced, without much of an effort of providing base data that was used for further analysis.

Reviewer 4 Report

It's an interesting topic. However, the approach is less structured. It is necessary to specify the purpose and objectives of the study, in detail. They (both the purpose and the objectives) are captured in the content, but do not provide visibility to the research. They must be mentioned at the beginning and followed in the results.

Summary intervention needed. Through its structure, it must include the description of the material to be presented. It is an ambiguous presentation.

In the Introduction, several citations are needed, which may be necessary to support and support the ideas mentioned.

It is necessary to standardize the writing, taking into account that it is an academic work - Fig. 4.

The results of the study are not mentioned as part of the research carried out. They are supplemented with other structural elements, but they do not provide that clarity in the pursuit of ideas - purpose.